# Energy cost of entanglement extraction in complex quantum systems

Cédric Bény[1], Christopher T. Chubb [2], Terry Farrelly[3] & Tobias J. Osborne[3]

What is the energy cost of extracting entanglement from complex quantum systems? Operationally, we may wish to actually extract entanglement. Conceptually, we may wish to physically understand the entanglement distribution as a function of energy. This is important, especially for quantum field theory vacua, which are extremely entangled. Here we build a theory to understand the energy cost of entanglement extraction. First, we consider a toy model, and then we define the entanglement temperature, relating energy cost to extracted entanglement. Next, we give a physical argument quantifying the energy cost of entanglement extraction in some quantum field vacua. There the energy cost depends on the spatial dimension: in one dimension, for example, it grows exponentially with extracted entanglement. Next, we provide approaches to bound the energy cost of extracting entanglement more generally. Finally, we look at spin chain models numerically to calculate the entanglement temperature using matrix product states.

[1] Department of Applied Mathematics, Hanyang University (ERICA), 55 Hanyangdaehak-ro, Ansan, Gyeonggi-do 426-791, Korea. [2] Centre for Engineered Quantum Systems, School of Physics, University of Sydney, Sydney, NSW 2006, Australia. [3] Institut für Theoretische Physik, Leibniz Universität Hannover, Appelstr. 2, 30167 Hannover, Germany. Correspondence and requests for materials should be addressed to T.F. (email: farreltc@tcd.ie)

Entanglement is a key resource in quantum information, with a vast array of practical uses[1]. In physics more generally, understanding the entanglement structure in physical systems is highly illuminating, e.g., in condensed matter theory where quantum phase transitions are signalled by long-range entanglement[2–6]. In high energy physics, quantifying the entanglement of states of a quantum field has applications to a variety of problems[7,8], from the AdS/CFT correspondence[9], through to detecting spacetime curvature by probing vacuum entanglement[10] or harvesting this entanglement by locally coupling small systems to the field[11–18].

Quantifying entanglement in states of quantum fields is a nontrivial task due to the UV dependence of quantum entanglement near a boundary, leading to naive divergences[19–21]. Many ad hoc approaches have been developed to deal with these divergences, usually relying on subtracting the UV divergent piece[19]. Of course, operationally, there are no such divergences in the entanglement we can measure because any apparatus we can build to extract entanglement from the field vacuum would only use a finite amount of energy.

Surprisingly little work has been done in the quantum information literature on the problem of quantifying accessible entanglement subject to an energy constraint, or, similarly, quantifying how much energy it costs to extract entanglement from a quantum system. There are some very interesting related ideas in the literature: in ref. [22], general quantum operations costing zero energy are studied. Also, the energy cost of creating entanglement in specific many-body systems was calculated in ref. [23]. Similarly, in the setting of quantum thermodynamics, the energy/work cost of creating correlations in quantum systems was studied in refs. [24–26]. These give useful strategies for creating correlations between finite dimensional systems or a pair of bosons or fermions using energy-conserving (global) unitary operations in the presence of heat baths. In ref. [27], entanglement distillation is considered (also in the presence of a heat bath) with an energy constraint: asymptotically many entangled pairs are distilled into EPR pairs, with the constraint that the energy before and after is equal. In ref. [28], using a specific local entanglement harvesting protocol (called entanglement farming), the energy cost in the low energy regime was calculated. In contrast, one may be interested in how the optimal energy cost scales with the number of EPR pairs extracted and in the overall entanglement structure of states, which is a rather different question.

Here we introduce a theory for the energy cost of entanglement extraction via local operations and classical communication (LOCC), focussing mostly on ground states. We use the term extraction rather than one-shot distillation[29], as we want to emphasize that we are not necessarily distiling all the entanglement from a state. In contrast, we wish to quantify the optimal energy cost per EPR pair extracted. While individual protocols for entanglement extraction are interesting, we are primarily concerned with the protocol that minimises the energy cost. So one of the benefits of our results is that we focus on protocol-independent lower bounds on the energy cost of actually extracting useful entanglement from complex systems. Another benefit is that this elucidates the entanglement structure of complex systems from an operational perspective: we gain an understanding of how useful (meaning extractable or distillable) entanglement is distributed in many-body or field systems as a function of the energy cost of actually accessing it. Furthermore, this is timely, as recent insights in high energy physics and gravity have connected spacetime structure to entanglement[30,31]. We first study the energy cost of entanglement extraction using a toy model, which is chosen to share many of the features of the vacuum state of a quantum field. Next, we introduce the entanglement temperature, which relates the amount of entanglement

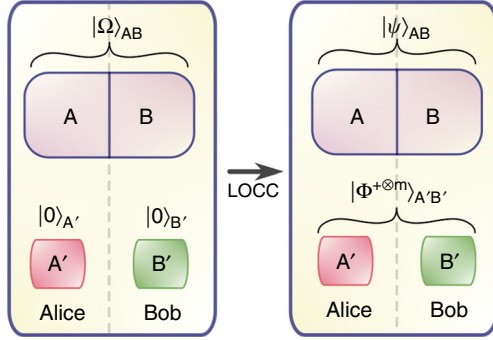

**Fig. 1** Energy cost of entanglement extraction: Alice and Bob have access to two parts of a quantum system in an entangled state $|\Omega\rangle_{AB}$ with Hamiltonian $H_{AB}$. Using local operations and classical communication (LOCC), they extract $m$ EPR pairs into their ancillary systems A′ and B′, leaving the physical system in the final state $|\psi\rangle_{AB}$. Dropping subscripts, the energy cost is then $\Delta E = \langle\psi|H|\psi\rangle - \langle\Omega|H|\Omega\rangle$. Of primary interest to us is the minimal energy cost, where Alice and Bob perform such an extraction protocol outputting $m$ EPR pairs with the lowest value for $\Delta E$

extracted to the energy cost. Then, we look at the energy cost of entanglement extraction in quantum field theories using physical arguments. We also discuss some general methods for quantifying the energy cost of entanglement extraction. Finally, we use matrix product states to numerically calculate the energy cost of entanglement extraction for some condensed matter systems.

## Results
**Our setting**. First, we focus on a simplified setup (see Fig. 1) exemplifying the core features of our problem: Alice and Bob have access to a bipartition of a common system with Hilbert space $\mathcal{H}_{AB}$. This system, which we refer to as the physical system, has a Hamiltonian $H_{AB}$, which neither Alice nor Bob can modify. Alice and Bob also have access to local ancillary degrees of freedom A′B′, which they can use to store the entanglement they extract from the physical system. Thus, the total Hilbert space for the system is given by

$$\mathcal{H}_{AA'BB'} \equiv \mathcal{H}_{AB} \otimes \mathcal{H}_{A'B'}. \tag{1}$$

We assume that Alice and Bob can carry out any local operation they like on the ancillary degrees of freedom with no energy cost. Thus we associate to the total system + ancilla system the Hamiltonian $H = H_{AB} \otimes 1_{A'B'}$.

We assume that Alice and Bob can only perform local operations and classical communication (LOCC)[32], which is natural, as this includes all operations distant parties can do in practice, and prevents them from cheating by, e.g., simply creating entanglement. We also suppose that Alice and Bob are working in the one-shot regime, which is natural if, for example, we are thinking of understanding the entanglement structure of vacuum states in quantum field theory where there is only one copy of the system available. In contrast, in the asymptotic many-copy regime we could use entanglement distillation protocols[1]. We will also comment on the energy cost in the asymptotic regime.

We quantify the energy cost as follows. Suppose we have a completely positive trace-preserving map (also called a quantum channel) $\mathcal{E} : \mathcal{E}(\mathcal{H}) \to \mathcal{S}(\mathcal{H})$ acting on the space $\mathcal{S}(\mathcal{H})$ of density operators on a Hilbert space $\mathcal{H}$. These are the most general operations one can do in quantum theory, and include measurements as well as applying unitaries to systems[32]. Suppose we also have a Hamiltonian $H \in \mathcal{B}(\mathcal{H})$. Given a state $\rho \in \mathcal{S}(\mathcal{H})$,

the operation $\mathcal{E}$ induces an energy change

$$\Delta(\mathcal{E}, \rho) = \mathrm{tr}(\mathcal{E}(\rho)\mathrm{H}) - \mathrm{tr}(\rho\mathrm{H}) \qquad (2)$$

when acting on $\rho$ (this can be negative). This is the energy cost when we apply the channel $\mathcal{E}$ to the state $\rho$. As an aside, we may also define the energy cost for the channel $\mathcal{E}$, defined by the largest possible change in energy after application of $\mathcal{E}$:

$$
\begin{aligned}
\Delta(\mathcal{E}) &= \sup_{\rho \,\in\, \mathcal{S}(\mathcal{H})} \mathrm{tr}(\mathcal{E}(\rho)\mathrm{H}) - \mathrm{tr}(\rho\mathrm{H}) \\
&= \|\mathrm{H} - \mathcal{E}^*(\mathrm{H})\|_\infty .
\end{aligned}
\qquad (3)
$$

where $\|\cdot\|_\infty$ is the operator norm and $\mathcal{E}^*$ is the dual of $\mathcal{E}$ in the Heisenberg picture.

**Extracting entanglement subject to an energy constraint.** Here we propose a definition for the entanglement accessible to Alice and Bob when they have access only to operations costing less than $\Delta E$.

We imagine that the system AB starts in a state $\sigma_{\mathrm{ABA'B'}} = |\Omega\rangle_{\mathrm{AB}}\Omega \otimes |00\rangle_{\mathrm{A'B'}}00$ where $|0\rangle$ is a convenient fiducial state of the ancilla and $|\Omega\rangle_{\mathrm{AB}}$ is the initial state of the physical system. Alice and Bob are now allowed to carry out LOCC operations costing less than $\Delta E$ in total to maximise the quantum entanglement between A′ and B′. This is the extraction step. Alice and Bob take some entanglement from the AB system and convert it into some number of maximally entangled EPR pairs, e.g., $m$ copies of the state $|\Phi^+\rangle = \frac{1}{\sqrt{2}}(|00\rangle + |11\rangle)$, which is a maximally entangled state of two qubits.

Suppose that $\mathcal{E}$ is a LOCC quantum channel and define $\rho_{\mathrm{ABA'B'}} = \mathcal{E}(\sigma_{\mathrm{ABA'B'}})$. We define the energy cost of extracting $m$ EPR pairs to be

$$\Delta E = \min \Delta(\mathcal{E}, \sigma_{\mathrm{ABA'B'}}), \qquad (4)$$

where the minimum is over all LOCC channels satisfying $\rho_{\mathrm{A'B'}} = |\Phi^{+\otimes m}\rangle_{\mathrm{A'B'}}\langle\Phi^{+\otimes m}|$.

From a different point of view, one can also define the entanglement accessible with energy $\Delta E$ to be

$$\mathrm{Ent}_{\Delta E}(|\Omega_{\mathrm{AB}}\rangle) \equiv \sup_{\Delta(\mathcal{E}, \sigma_{\mathrm{ABA'B'}}) \,\leq\, \Delta E} \mathrm{Ent}(\rho_{\mathrm{A'B'}}), \qquad (5)$$

where again $\mathcal{E}$ is a LOCC channel and Ent is some convenient entanglement measure. For pure states, one usually takes the entanglement entropy, but for mixed states the entanglement entropy is not a sensible entanglement measure, and one must choose one of several different measures quantifying mixed-state entanglement[1]. Here we usually deal with pure states, so we will indeed use the entanglement entropy, which is defined by $\mathrm{Ent}(\rho_{\mathrm{AB}}) = S(\rho_{\mathrm{B}}) = -\mathrm{tr}[\rho_{\mathrm{B}}\log_2(\rho_{\mathrm{B}})]$, where $\rho_{\mathrm{B}} = \mathrm{tr}_{\mathrm{A}}[\rho_{\mathrm{AB}}]$.

It is conceivable that after extracting entanglement, the energy of the system can go down, i.e., as well as extracting entanglement, Alice and Bob may extract some energy. This depends on the state $|\Omega_{\mathrm{AB}}\rangle$, i.e., whether it is an excited state or ground state. Since the emphasis in this paper is on ground states, we assume henceforth that $|\Omega_{\mathrm{AB}}\rangle$ is the ground state of $\mathrm{H}_{\mathrm{AB}}$.

One can generalize the scenario above to give a more comprehensive account of the energy cost. Consider the following situation: the total system comprises (i) the physical system plus Alice and Bob's ancillas (now with nontrivial Hamiltonians $\mathrm{H}_{\mathrm{A'}}$ and $\mathrm{H}_{\mathrm{B'}}$ respectively), (ii) batteries which allow us to account for any energy that Alice and Bob use to perform their operations with Hamiltonian $\mathrm{H}_{\mathrm{batt}}$, and (iii) the environment which has Hamiltonian $\mathrm{H}_{\mathrm{en}}$. We can model whatever operations Alice and Bob do on the total system by a unitary or quantum channel with

the sole demand that it conserves the total energy. (Any energy they need to actually extract entanglement comes from the battery system.) Then we have that the total Hamiltonian is $\mathrm{H} = \mathrm{H}_{\mathrm{AB}} + \mathrm{H}_{\mathrm{A'}} + \mathrm{H}_{\mathrm{B'}} + \mathrm{H}_{\mathrm{batt}} + \mathrm{H}_{\mathrm{en}}$, and energy conservation implies that $\mathrm{tr}[\rho_{\mathrm{I}}\mathrm{H}] = \mathrm{tr}[\rho_{\mathrm{F}}\mathrm{H}]$, where $\rho_{\mathrm{I}}$ and $\rho_{\mathrm{F}}$ are the initial and final states of the whole system, respectively. As the energy cost from the perspective of the experimentalists (Alice and Bob) is $\Delta E = \mathrm{tr}[(\rho_{\mathrm{I}} - \rho_{\mathrm{F}})\mathrm{H}_{\mathrm{batt}}]$, we get

$$\Delta E = \mathrm{tr}[(\rho_{\mathrm{F}} - \rho_{\mathrm{I}})(\mathrm{H}_{\mathrm{AB}} + \mathrm{H}_{\mathrm{A'}} + \mathrm{H}_{\mathrm{B'}} + \mathrm{H}_{\mathrm{en}})]. \qquad (6)$$

Now, we assume that $\mathrm{tr}[(\rho_{\mathrm{F}} - \rho_{\mathrm{I}})\mathrm{H}_{\mathrm{en}}] \geq 0$, otherwise they are getting energy from the environment, which should be accounted for by the battery. Then we get

$$\Delta E \geq \mathrm{tr}\big[(\rho_{\mathrm{F}} - \rho_{\mathrm{I}})(\mathrm{H}_{\mathrm{AB}} + \mathrm{H}_{\mathrm{A'}} + \mathrm{H}_{\mathrm{B'}})\big]. \qquad (7)$$

Now for the setting we are interested in, we assume Alice and Bob's ancillas are well understood systems, which can be finely controlled (after all, they may want to use the entanglement they extract for some useful task, such as teleportation). Because of this, one should expect to be able to easily calculate $\mathrm{tr}[(\rho_{\mathrm{F}} - \rho_{\mathrm{I}})(\mathrm{H}_{\mathrm{A'}} + \mathrm{H}_{\mathrm{B'}})]$, as the initial and final states of their ancillas are known: they prepare the ancillas in some initial state and then they extract some number of EPR pairs. (It is also perfectly reasonable to assume that the ancillas are initially in their ground states.) Therefore, the interesting calculation is finding $\mathrm{tr}[(\rho_{\mathrm{F}} - \rho_{\mathrm{I}})\mathrm{H}_{\mathrm{AB}}]$. So it is reasonable to focus on bounding this term below, as this is independent of the setup of Alice and Bob's ancillas, which, even if nontrivial, would be simple in any reasonable protocol (e.g., some qubits where the $i$th qubit has Hamiltonian $\mathrm{H}_i \propto \sigma_z$).

Furthermore, of course, one would also be interested in very precise energy accounting for practical purposes, but the energy cost of changing the state of the physical system due to extracting entanglement will always be present and is independent of the setup and therefore fundamental.

**Weak vs strong interactions.** A key ingredient in any entanglement extraction protocol is the strength of the interaction between Alice's and Bob's systems. If we write $\mathrm{H}_{\mathrm{AB}} = \mathrm{H}_{\mathrm{A}} \otimes \mathbb{1}_{\mathrm{B}} + \mathbb{1}_{\mathrm{A}} \otimes \mathrm{H}_{\mathrm{B}} + V_{\mathrm{AB}}$, then the limitations on how much entanglement Alice and Bob can extract using LOCC are determined by $V_{\mathrm{AB}}$. Indeed, if $V_{\mathrm{AB}} = 0$, the ground state $|\Omega\rangle$ will have no entanglement between Alice's and Bob's systems.

There is a useful naive protocol for entanglement extraction: Alice and Bob first swap the states of their primed and non-primed systems. Then they can prepare a state of the physical AB system (using LOCC) with minimal local energy, meaning Alice/Bob prepares $|\psi_{\mathrm{A/B}}\rangle$, such that $\langle\psi_{\mathrm{A/B}}|\mathrm{H}_{\mathrm{A/B}}|\psi_{\mathrm{A/B}}\rangle$ is minimised. The total energy change is, with $|\psi\rangle = |\psi_{\mathrm{A}}\rangle|\psi_{\mathrm{B}}\rangle$,

$$
\begin{aligned}
&\langle\psi|\mathrm{H}|\psi\rangle - \langle\Omega|\mathrm{H}|\Omega\rangle \\
&\leq \langle\psi|V_{\mathrm{AB}}|\psi\rangle - \langle\Omega|V_{\mathrm{AB}}|\Omega\rangle \leq 2\|V_{\mathrm{AB}}\|_\infty .
\end{aligned}
\qquad (8)
$$

Therefore, when the coupling is sufficiently weak, Alice and Bob can safely extract all the entanglement whilst only incurring a small energy cost.

In contrast, for strong couplings the situation is entirely different, which is exactly the case for quantum field theories, where extracting all the entanglement costs a divergent amount of energy. For an example of a free fermion field, we see in the Methods section that all product states $|\psi\rangle$ satisfy $\langle\psi|\mathrm{H}|\psi\rangle \geq 1/a$, where $a$ is the regulator (the lattice spacing in this case). Thus, the energy diverges as $a \to 0$ for any product state, meaning that

extracting all the entanglement costs a diverging amount of energy. In general, the energy cost for extracting all the entanglement will diverge in quantum field theory. Again using a lattice regulator, the energy contained in the interaction terms between a region A and the rest scales like $(\partial A/a^d)$, where $\partial A$ is the boundary of A, which is also shown in the Methods section.

**A toy model**. In this section we discuss an idealised model, which exemplifies many of the features of the quantum field vacuum. It has high entanglement and a high energy cost for extracting all this entanglement, as we will see.

Suppose that the system AB is actually composed of $2n$ qubits, with $n$ qubits in A and $n$ qubits in B. We call the qubits $A_j$ and $B_j$, respectively, for $j = 1, 2, \ldots, n$. We suppose that $H_{AB}$ is given by

$$H_{AB} = \sum_{j=1}^{n} \left(1 - P_{A_j B_j}\right), \tag{9}$$

where $P_{A_j B_j}$ is the projector onto the maximally entangled state $|\Phi^+\rangle = \frac{1}{\sqrt{2}}(|00\rangle + |11\rangle)$ of qubits $A_j$ and $B_j$. The ground state $|\Omega_{AB}\rangle$ of $H_{AB}$ is thus a product of maximally entangled pairs, i.e. it is a maximally entangled state between A and B.

If Alice and Bob could do arbitrary LOCC, then they could easily extract $n$ EPR pairs. However, if they are only allowed an energy cost of $\Delta E$, then naively they should only be able to extract $O(\Delta E)$ EPR pairs.

In the most extreme case, Alice and Bob fully extract all the EPR pairs. Then, in order that this entanglement is between ancilla degrees of freedom in A′ and B′, it must be that A and B are in a separable state $\sigma_{AB}$. Since the energy depends linearly on $\sigma_{AB}$, we may as well suppose that $\sigma_{AB}$ is an extreme point of the convex set of separable states, namely, a product state $|\phi\rangle_A |\psi\rangle_B$. The energy of our initial state $|\Omega\rangle_{AB}$ was zero, so the energy cost of any entanglement extraction procedure must be greater than

$$\inf_{|\phi\rangle_A |\psi\rangle_B} \sum_{j=1}^{n} \left(1 - \langle\phi_A|\langle\psi_B|P_{A_j B_j}|\phi_A\rangle|\psi_B\rangle\right). \tag{10}$$

This infimum is achieved by finding the supremum:

$$\sup_{|\phi\rangle_A |\psi\rangle_B} \langle\phi_A|\langle\psi_B|P_{A_j B_j}|\phi_A\rangle|\psi_B\rangle, \tag{11}$$

which is equal to 1/2. (e.g. setting each pair to $|00\rangle_{A_i B_i}$ will do the job.) Thus, the energy cost is given by

$$\Delta E \geq \frac{1}{2}\sum_{j=1}^{n} 1 = n/2. \tag{12}$$

More generally, suppose Alice and Bob extract fewer EPR pairs (say $m$), one option is to use the following simple protocol. They swap the states of the first $m$ EPR pairs of the physical system into their ancilla systems, which they can do using local operations. The first $m$ pairs of qubits of the physical system are now in a product pure state. Then they can apply local unitaries mapping each of these qubit pairs to the state $|00\rangle_{A_i B_i}$, getting the final energy cost

$$\Delta E = \frac{1}{2}\sum_{j=1}^{m} 1 = m/2. \tag{13}$$

Thus, the total energy cost is 1/2 per EPR pair extracted.

Of course, there may be a protocol extracting the same amount of entanglement but costing less energy. In the Methods section,

we argue that the simple protocol given above is, in fact, optimal. The argument relies on a combination of numerics and the majorization criteria, which determine when pure state transformations using LOCC are possible. To introduce majorization, we use the Schmidt decomposition: for any pure state $|\phi\rangle_{AB}$, there exist orthonormal bases of A and B, denoted $|a_i\rangle$ and $|b_i\rangle$ respectively, such that $|\phi\rangle_{AB} = \sum_{i=1}^{2^n} \alpha_i |a_i\rangle|b_i\rangle$, where $\alpha_i$ are called Schmidt values and are positive real numbers decreasingly ordered[32]. For it to be possible to transform this state into the new state $|\psi\rangle_{AB}$, with Schmidt values $\beta_i$, the majorization condition[32] must be satisfied:

$$\forall K \geq 1 \quad \sum_{i=1}^{K}|\alpha_i|^2 \leq \sum_{i=1}^{K}|\beta_i|^2. \tag{14}$$

For this to be possible, the Schmidt rank of the resulting state (the number of non zero $\beta_i$) must be smaller than that of the initial state. The LOCC protocol implementing such transformations can be found in ref. [32].

On the other hand, in the asymptotic entanglement distillation setting, the energy cost above for the toy model is not optimal. In that case, one can distil entanglement at a lower energy cost, which we show in the Methods section. In practice, however, we only have access to one copy of a quantum field or condensed matter system, so it is crucial to consider the one-shot setting. Furthermore, entanglement distillation protocols rely on projecting onto typical subspaces defined by the Schmidt vectors of the initial state[32], which for extremely complex systems would be practically impossible.

**The entanglement temperature**. In the previous section, the total energy cost was 1/2 per EPR pair extracted. To relate the change in entanglement entropy $\Delta S$ to the energy cost $\Delta E$, we define the entanglement temperature $T_{ent}$ by

$$\Delta E = T_{ent}\Delta S. \tag{15}$$

(The name entanglement temperature is chosen in analogy with thermodynamics, although it is important to emphasize that there is generally no connection with thermodynamic temperature.) So $T_{ent}$ is a property of the ground state of a system. For the toy model, we see that $T_{ent} = 1/2$ since $\Delta S = m\log_2(2) = m$. In this case $T_{ent}$ is constant because there is a linear relationship between the entanglement extracted and the energy cost. For general systems, we would not expect $\Delta E \propto \Delta S$ for the entire range of $\Delta S$. Instead, we should think of the entanglement temperature as a function of the extracted entanglement. (This is also true in thermodynamics, where thermodynamic temperature can often be thought of as a function of other state functions, such as entropy or pressure.)

In the following sections, we give some physical and numerical arguments to find $\Delta E$ as a function of $\Delta S$ and hence find the entanglement temperature for some physical systems.

**The energy cost in quantum field theory**. For some quantum field theories, we can give a physical argument for the energy cost of entanglement extraction. Using known formulas for the entropy of ground states of lattice models[33], we have an expression for the entropy of certain QFTs regulated on a lattice with lattice spacing $a$. By estimating the energy cost of extracting all the entanglement from this regulated vacuum state, we get an expression for the energy cost as a function of the lattice spacing. Combining these formulae, we find an estimate for the energy cost of extracting a given amount of entanglement.

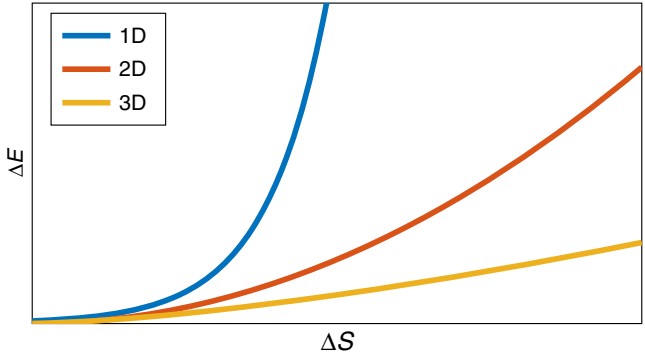

**Fig. 2** Energy cost of entanglement extraction from quantum field vacua: this depends heavily on the spatial dimension. Here we sketch the behaviour in dimensions $d = 1, 2, 3$. When $d = 1$, $\Delta E \propto \exp(K\Delta S)$, where $K$ is a constant, and for $d > 1$, $\Delta E \propto \Delta S^{d/(d-1)}$

For example, for one dimensional lattice systems, the entanglement entropy in the ground state of models close to the critical point is[20,33]

$$S(\rho_{\mathrm{I}}) = \frac{c}{6} \log_2(\xi/a), \tag{16}$$

where $\xi \gg a$ is the correlation length, $c$ is a constant and $\rho_{\mathrm{I}}$ is the state corresponding to the infinite half-line $(-\infty, 0]$. This is equivalent to a massive relativistic QFT with $1/\xi$ equal to the mass, e.g., for free bosons we have $c = 1$.

As argued in the section "Weak vs strong interactions", the energy cost for extracting all of this entanglement is $\Delta E = O(1/a)$. (In some cases, exactly half of this entanglement is one-shot distillable[34].) Supposing we wished to extract $m$ EPR pairs, then we need to probe an energy scale such that $S(\rho_{\mathrm{I}}) = \frac{c}{6} \log_2(\xi/a) \geq m$, which corresponds to an energy cost of at least $1/a$ where $a = \xi/2^{-6m/c}$. Therefore, we have the energy cost $\Delta E \propto \exp(Km)$, where $K = 6\ln(2)/c$. This means that the energy cost of entanglement extraction increases exponentially: there is infinite entanglement in the quantum field vacuum, but the cost of extraction grows quickly. (Note that in practice Alice and Bob would only have access to finite regions of space, so this actually gives a lower bound on the energy cost in such scenarios.) For gapless (i.e. critical) models, the same argument goes through if Alice has access to a finite region and Bob has access to the rest. In contrast, if Alice's system is a half-line, there is infinite entanglement at any energy scale. One can see this using the formula for the entanglement entropy at criticality between the state of the lattice system $\rho_{\mathrm{I}}$ on an interval I of length $l$ and the rest, given by $S(\rho_{\mathrm{I}}) = \frac{c}{3} \log_2(l/a) + c'$, where $c$ and $c'$ are constants[33].

The scaling is different for quantum fields in higher dimensional spaces. In many cases, e.g. free massive quantum fields[20], the entanglement entropy of the ground state is known to obey an area law[35]. Then in a region A with area $\partial A$, the leading contribution to the entropy is $S(\rho_A) \propto \partial A/a^{d-1}$[20]. However, the energy cost of extracting all the entanglement scales like $\Delta E \propto \partial A/a^d$ (This is justified in the Methods section). Thus we arrive at an estimate for the energy cost of entanglement extraction: $\Delta E \propto \Delta S^{d/(d-1)}$. And the entanglement temperature is then $T_{\mathrm{ent}} \propto 1/a \propto \Delta E^{1/d}$. The energy costs of entanglement extraction in QFTs are plotted in Fig. 2.

These arguments work for quantum field theories where the entanglement entropy of the ground state of the model (regulated on a lattice) can be calculated. This works well for free field theories, though it is believed that entanglement area laws hold more generally, so the ideas should extend further.

For more general systems, there is no clear way to proceed. Here we outline some potential methods to approach the problem (including optimization via Lagrange multipliers in the Methods section), and we consider the trade-off between entanglement change and energy cost numerically.

**Lower bound for gapped systems**. One option is to maximise the overlap of the final state of the physical system (after the entanglement has been extracted) with its ground state. For Hamiltonians of the form $H_{\mathrm{AB}} = -|\Omega\rangle\langle\Omega|$, we get an exact answer for the optimal energy cost. Also, one can use this method to get a lower bound on the energy cost for a Hamiltonian with a gap $\Delta$ and a non-degenerate ground state as follows. Suppose the final state is $|\psi\rangle$, then the energy cost is $\langle\psi|H_{\mathrm{AB}}|\psi\rangle$, assuming the ground state energy is zero, without loss of generality. Then we get

$$\begin{aligned}
\langle\psi|H_{\mathrm{AB}}|\psi\rangle &= \langle\psi|H_{\mathrm{AB}}(1 - |\Omega\rangle\langle\Omega|)|\psi\rangle \\
&\geq \Delta\langle\psi|(1 - |\Omega\rangle\langle\Omega|)|\psi\rangle \\
&= \Delta\left(1 - |\langle\Omega|\psi\rangle|^2\right).
\end{aligned} \tag{17}$$

As an example, take $|\Omega\rangle = (1/\sqrt{d}) \sum_{i=1}^{d} |i\rangle_{\mathrm{A}}|i\rangle_{\mathrm{B}}$, where $d = 2^n$. Suppose that Alice and Bob extract $m$ EPR pairs using LOCC, leaving a pure state $|\psi\rangle$ in the physical system. Using the majorization criterion, this can be any state with Schmidt rank up to $K = 2^{n-m}$. To minimise the energy cost, we need to find such a state having maximal overlap with $|\Omega\rangle$.

We may write the optimal $|\psi\rangle$ in its Schmidt basis as $\sum_{i=1}^{K} \alpha_i |a_i\rangle_{\mathrm{A}}|b_i\rangle_{\mathrm{B}}$. Next notice that

$$\left( \sum_{i=1}^{d} \langle i|_{\mathrm{A}}\langle i|_{\mathrm{B}} \right) \left|a_j\right\rangle_{\mathrm{A}} \left|b_j\right\rangle_{\mathrm{B}} \leq 1. \tag{18}$$

Then we have

$$\langle\Omega|\psi\rangle \leq \frac{1}{\sqrt{d}} \sum_{i=1}^{K} \alpha_i \leq \sqrt{\frac{K}{d}} = 2^{-m/2}. \tag{19}$$

Therefore, if the Hamiltonian for this system has gap $\Delta$, we see that the energy cost for extracting $m$ EPR pairs is at least $\Delta(1 - 2^{-m})$. (Applying this method to our earlier toy model actually gives a tight lower bound when one EPR pair is extracted.)

**Numerics**. Another option is to consider the trade-off between entanglement and energy numerically. For a given Hamiltonian H, we consider a procedure in which the system starts in the ground state $|\Omega\rangle$, some entanglement is extracted, and the system is left in a final state $|\psi\rangle$. The energy cost of this procedure is $\langle\psi|H|\psi\rangle - \langle\Omega|H|\Omega\rangle$, and the extracted entropy is upper bounded by $\mathrm{Ent}(|\Omega\rangle) - \mathrm{Ent}(|\psi\rangle)$. In the asymptotic many-copy case, this is exactly the extracted entanglement entropy. We denote the entanglement temperature in that case by $T_{\mathrm{ent}}^{\mathrm{A}}$. We have that

$$T_{\mathrm{ent}}^{\mathrm{A}} \leq \frac{\langle\psi|H|\psi\rangle - \langle\Omega|H|\Omega\rangle}{\mathrm{Ent}(\Omega) - \mathrm{Ent}(\psi)}. \tag{20}$$

Note that for a given amount of extracted entanglement $\Delta S$, the one-shot entanglement temperature is lower bounded by the asymptotic-setting entanglement temperature $T_{\mathrm{ent}}^{\mathrm{A}} \leq T_{\mathrm{ent}}$.

A given state does not necessarily give a tight bound on $T_{\mathrm{ent}}^{\mathrm{A}}$. For this we need to study the optimal trade-off between

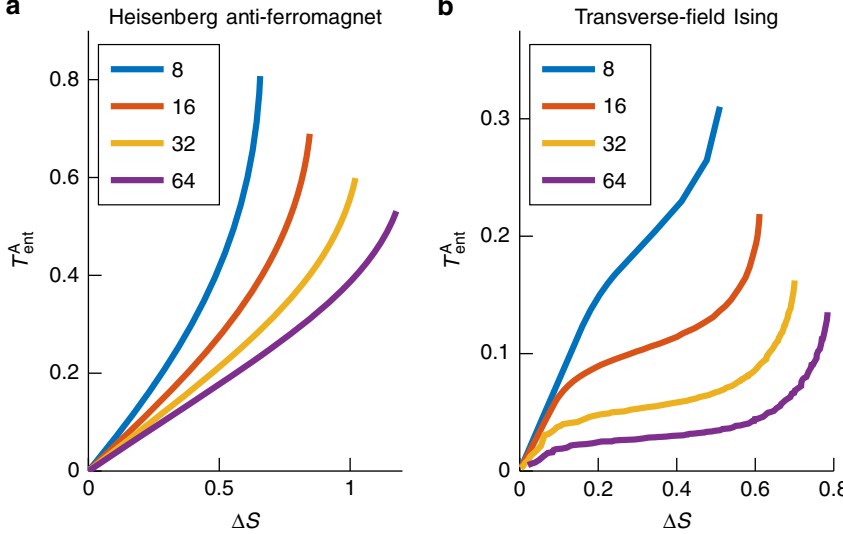

**Fig. 3** Asymptotic-case entanglement temperature for two critical spin models. $\left(T_{ent}^{A} = \Delta E/\Delta S\right)$ as a function of entropy change $\Delta S$ for the critical Heisenberg anti-ferromagnet (**a**) and the critical transverse-field Ising model (**b**), for multiple system sizes. Notice that near the ground state $T_{ent}^{A} \propto \Delta S$, which we prove is generic in the Methods section. The one-shot entanglement temperature $T_{ent}$ is lower bounded by the asymptotic temperature $T_{ent}^{A}$

entanglement and energy, which is given by a Pareto front. By randomly generating states with low energy and low entanglement, we can numerically evaluate the above upper bound, and use this to compute the Pareto front. We describe a tensor network method for generating such samples in the Methods section. In Fig. 3 we present numerical results for two 1D spin models.

It is interesting to note that the entanglement temperature for a fixed change in entropy (amount of entanglement extracted) actually decreases with increasing system size for the critical models studied in Fig. 3. There is a physical reason for this. At criticality, the ground state is scale-invariant, so there is entanglement at all length scales, so Alice and Bob might as well focus on just the low energy (large length-scale) modes. And for bigger and bigger systems the gap closes, so that Alice and Bob can extract a lot of entanglement out of the low energy modes with very little energy cost.

Furthermore, the apparent divergence of the entanglement temperatures in Fig. 3 is a finite-size effect. The total amount of extractable entanglement grows with the system size, while the energy cost of extracting all entanglement is bounded above (by the strength of the interaction between Alice and Bob's systems). Therefore, the maximum entanglement temperature for any system size is bounded (since $T_{ent} = \Delta E/\Delta S$).

## Discussion

We introduced a framework to understand and quantify the energy cost of extracting entanglement from complex quantum systems. After looking at a toy model, which illustrated the key concepts, we defined the entanglement temperature. Then we analysed the energy cost of entanglement extraction in quantum field theories, and we saw that the energy cost of extracting entanglement depends on the spatial dimension. Finally, we looked at some general methods to approach the problem, including numerical methods for lattice models.

Quantifying how much energy extracting $m$ EPR pairs costs in physical systems illuminates the entanglement structure of states, particularly ground states of, e.g. quantum fields. But it can also upper bound how efficient protocols such as entanglement harvesting can be. For general systems, the optimal strategy for entanglement extraction may be hard to find. Still, it is heartening

that, at least for quantum field theories, there is a relatively simple formula for the energy cost.

It would be interesting to combine the ideas here with those in ref. [36], where transformations between entangled states are considered using an additional resource: an entanglement battery. This is a reservoir from which entanglement may be taken or deposited to facilitate state transformations, which may be impossible otherwise. One may then ask how this theory changes when there is also an energy cost associated with using the entanglement in the battery.

Another interesting possibility is to go beyond ground states, considering perhaps excited or thermal states of the physical system. While the entanglement properties of such states are less well understood, they are also quite interesting: excited states are generally expected to have more entanglement than ground states, though thermal states are easier to prepare in practice.

## Methods

**Cost of extracting all correlations in lattice QFT.** Consider free massless fermions in $(1 + 1)$ dimensions with lattice cutoff $a$. To avoid worrying about fermion doubling, take staggered fermions[37], with Hamiltonian

$$H = \sum_{n=0}^{N-1} \left[ \frac{i\left(\psi_n^\dagger \psi_{n+1} - \psi_{n+1}^\dagger \psi_n\right)}{2a} + \frac{1}{a} \right], \tag{21}$$

where $\psi_n$ are fermion annihilation operators, satisfying $\{\psi_n, \psi_m^\dagger\} = \delta_{n,m}$ and $\{\psi_n, \psi_m\} = 0$. We have chosen the $1/a$ term on the right so that, as $a \to 0$, the Hamiltonian is positive definite with ground state energy independent of $a$ (which we verify at the end of the section). This normalisation also has the advantage that each term in the sum is positive definite:

$$i\left(\psi_n^\dagger \psi_{n+1} - \psi_{n+1}^\dagger \psi_n\right) + 2 = \left(\psi_n^\dagger - i\psi_{n+1}^\dagger\right)\left(\psi_n + i\psi_{n+1}\right) + \psi_n \psi_n^\dagger + \psi_{n+1}\psi_{n+1}^\dagger. \tag{22}$$

Now suppose Alice's system is one half of the chain (sites 0 to $N/2 - 1$), and Bob's system is the other half. If they extract the entanglement by swapping the state of the physical system into an ancilla, then the final state of the chain (which can be chosen to be pure because the energy can always be minimised by a pure state) is

$$|\psi\rangle = \left(\alpha\psi_{N/2-1}^\dagger A_1^\dagger + \beta A_2^\dagger\right)\left(\gamma\psi_{N/2}^\dagger B_1^\dagger + \delta B_2^\dagger\right)|0\rangle, \tag{23}$$

where $|0\rangle$ is the state satisfying $\psi_n|0\rangle = 0$ for all $n$; $A_1$, $A_2$ are products of annihilation operators on Alice's system, while $B_1$, $B_2$ are products of annihilation

operators on Bob's system; and $\alpha$, $\beta$, $\gamma$, and $\delta$ are complex numbers. Because of superselection rules, if $A_1$ is a product of an odd number of annihilation operators, then $A_2$ is even, or vice versa. The same holds for $B_1$ and $B_2$. Then one can easily verify that

$$\langle\psi|\left[\frac{i\left(\psi_n^\dagger\psi_{n+1}-\psi_{n+1}^\dagger\psi_n\right)}{2a}+\frac{1}{a}\right]|\psi\rangle\geq\frac{1}{a}. \tag{24}$$

So the energy of this state diverges as $a\to 0$.

Finally, we need to verify that we normalised the Hamiltonian suitably and that the ground state energy of the staggered-fermion Hamiltonian is independent of $a$. Thus, we need to diagonalise

$$H=\sum_{n=0}^{N-1}\left[\frac{i\left(\psi_n^\dagger\psi_{n+1}-\psi_{n+1}^\dagger\psi_n\right)}{2a}+\frac{1}{a}\right]. \tag{25}$$

To do this, we switch to momentum space, with

$$\psi_n=\frac{1}{\sqrt{N}}\sum_{k=0}^{N-1}e^{-2\pi ikn/N}\psi_k. \tag{26}$$

Then we have

$$H=\sum_{k=0}^{N-1}\left[-\frac{\sin(2\pi k/N)}{a}\psi_k^\dagger\psi_k+\frac{1}{a}\right]. \tag{27}$$

This has minimum energy

$$E_0=\frac{N}{a}-\frac{N}{2a}\left[\frac{2}{N}\sum_{k=0}^{N/2-1}\sin(2\pi k/N)\right], \tag{28}$$

where we have assumed that $N$ is even. We next define the Riemann sum over $[0,1]$ to be

$$R_M[f(x)]=\frac{1}{M}\sum_{m=1}^{M}f\left(\frac{m-1/2}{M}\right). \tag{29}$$

and we may use the following formula for convergence of a Riemann sum[38]

$$\left|R_M[f(x)]-\int_0^1 dx f(x)\right|\leq\frac{T(f')}{8M^2}, \tag{30}$$

which holds when $f$ is differentiable everywhere on $[0, 1]$ with bounded derivative and total variation $T(f')=\int_0^1 dx|f'(x)|$. To apply this to $E_0$, we write $M=N/2$. Then

$$\begin{aligned}\frac{1}{M}\sum_{k=0}^{M-1}\sin(\pi k/M)&=\frac{1}{M}\sum_{k=1}^{M}\sin(\pi(k-1)/M)\\&=\cos\left(\tfrac{\pi}{2M}\right)R_M[\sin(\pi x)]-\sin\left(\tfrac{\pi}{2M}\right)R_M[\cos(\pi x)]\\&=(1+O(M^{-2}))R_M[\sin(\pi x)]-O(M^{-1})R_M[\cos(\pi x)]\\&=R_M[\sin(\pi x)]+O(M^{-2})\\&=2+O(M^{-2}).\end{aligned} \tag{31}$$

To get the second line, we used that $\sin(A+B)=\sin(A)\cos(B)+\cos(A)\sin(B)$ and Eq. (29). In the third line, we used $\cos(x)=1+O(x^2)$ and $\sin(x)=O(x)$ for sufficiently small $x$. To get the fourth line, we used Eq. (30) to get $R_M[\cos(\pi x)]=0+O(M^{-2})$. And we used Eq. (30) again to get the last line.

Therefore, the ground state energy is

$$E_0=\frac{N}{a}-\frac{N}{2a}\left[2+O(N^{-2})\right]=O\left(\frac{1}{Na}\right), \tag{32}$$

and since $1/Na=1/L$ is a constant, $E_0$ is essentially independent of the lattice spacing.

For general lattice regularizations of quantum field theory, the energy contained in the interactions between a region A and the rest of the lattice scales like $(\partial A/a^{d-1})\times 1/a$, where $\partial A$ is the boundary of A. To see this, note that the region A has approximately $\partial A/a^{d-1}$ interaction terms with the rest, and the strength of each interaction is proportional to $1/a$. Dimensional analysis is often enough to argue this, but one can check for, e.g. massless scalar field theory in $d$ dimensions. In that

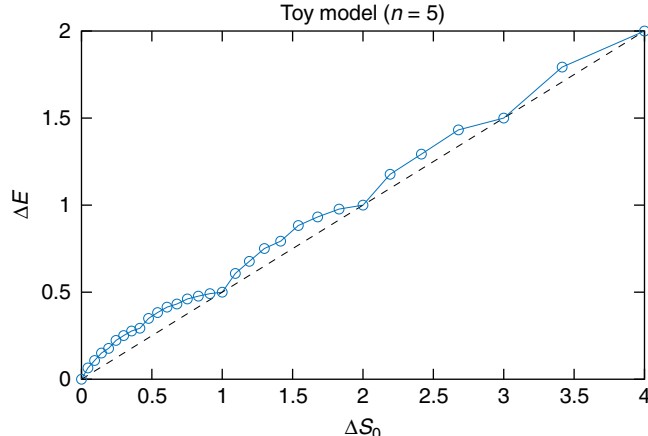

**Fig. 4** Minimum energy cost of extracting entanglement for the toy model. More precisely, the figure shows the minimum change in energy $\Delta E$ when there is a decrease in zero entropy $\Delta S_0$ (with $S_0$ equal to $\log_2$ of the Schmidt rank) of the state of the physical system. When $\Delta S_0$ is an integer $m$ (which corresponds to extracting $m$ EPR pairs from the system), then the plot shows $\Delta E=0.5m$. The calculation was performed using DMRG by restricting the bond dimension between Alice and Bob's systems

case, the interaction term between lattice sites comes from the discrete derivative:

$$(\nabla_a\phi)^2(\mathbf{n})=\sum_{i=1}^{d}\left(\frac{\phi(\mathbf{n}+a\mathbf{e}_i)-\phi(\mathbf{n})}{a}\right)^2, \tag{33}$$

where $\mathbf{e}_i$ are lattice basis vectors. The Hamiltonian is given by

$$H=\sum_{\mathbf{n}}\frac{a^d}{2}\left[\pi^2(\mathbf{n})+(\nabla_a\phi)^2(\mathbf{n})\right], \tag{34}$$

where $\pi(\mathbf{n})$ is the operator canonically conjugate to $\phi(\mathbf{n})$. Because $\phi(\mathbf{n})$ has dimensions of $(\text{length})^{(1-d)/2}$, we see that the interaction terms between sites have strength $O(1/a)$. One can also show that the energy density of a product state in quantum field theory is infinite[39].

**Optimality of the single-shot toy model protocol.** After applying their operations, Alice and Bob have $m$ Bell states in the ancilla $|\Phi^{+\otimes m}\rangle_{A'B'}$ and some pure state in the physical system $|\psi\rangle_{AB}$. Here we can use the majorization criterion. Writing $|\Omega\rangle_{AB}$ in its Schmidt basis (orthonormal bases of A and B, denoted $|a_i\rangle$ and $|b_i\rangle$ respectively), we have $|\Omega\rangle_{AB}=\sum_{i=1}^{2^n}\alpha_i|a_i\rangle|b_i\rangle$, where $\alpha_i$ are positive real numbers decreasingly ordered[32]. The initial state $|\Omega\rangle_{AB}|00\rangle_{A'B'}$ then has the same Schmidt rank as $|\Omega\rangle_{AB}$, which is $2^n$ in this case. For it to be possible to transform this state into the new state $|\psi\rangle_{AB}|\Phi^{+\otimes m}\rangle_{A'B'}$, the Schmidt rank of $|\psi\rangle_{AB}|\Phi^{+\otimes m}\rangle_{A'B'}$ must be smaller than that of $|\Omega\rangle_{AB}|00\rangle_{A'B'}$. This implies that the Schmidt rank of the new state of the physical system $|\psi\rangle_{AB}$ can be at most $2^{n-m}$.

Figure 4 shows numerics from a DMRG calculation of the minimum increase in energy as the Schmidt rank of the state of the physical system decreases. Based on these numerical results, we see that the minimum increase in energy when the Schmidt rank decreases by a factor of $2^m$ is 0.5 m. This can be achieved by the simple protocol of the previous section, indicating that this protocol is optimal. Actually, this whole argument also goes through even if Alice and Bob have some additional shared entanglement that can be used as a catalyst, as in ref. [40].

**Optimization via Lagrange multipliers.** One way to find the energy cost of extracting some entanglement is by finding the state (or set of states) of the physical system that has a given value of entanglement entropy with minimal energy. The idea is that, after Alice and Bob have extracted some entanglement, the lowest energy cost corresponds to leaving the system in such a state. For simplicity, we focus on finding a pure state $|\psi\rangle_{AB}$ with this property. (It is possible that a protocol giving a mixed state on the physical system may be more efficient. In this case, we may use a superadditive entanglement measure, like the squashed entanglement[41], to upper bound the entanglement left in the physical system.)

Alice and Bob have the initial state $|\Omega\rangle_{AB}|00\rangle_{A'B'}$, and then they apply some LOCC protocol to extract $m$ EPR pairs into the ancilla A'B'. Because they are using

LOCC, the overall entanglement can only decrease:

$$\text{Ent}\big(|\psi\rangle_{AB}|\Phi\rangle_{A'B'}^{+\otimes m}\big) \leq \text{Ent}\big(|\Omega\rangle_{AB}|00\rangle_{A'B'}\big) \qquad (35)$$
$$= S_{\text{initial}},$$

where $S_{\text{initial}}$ is the initial entanglement entropy in the state $|\Omega\rangle_{AB}$ and Ent is an entanglement measure, which we take to be the entanglement entropy, since the states are all pure. Then we have that the entanglement entropy in the final state of the physical system is $\text{Ent}(|\psi\rangle_{AB}) \leq S_{\text{initial}} - m$.

So we wish to minimise the energy, given that the entanglement entropy of the physical system is fixed. We can do this using Lagrange multipliers (analogously to how one derives the thermal state by maximising the entropy at fixed energy; see also ref. [42] for a similar calculation). Thus, we have the Lagrangian

$$\mathcal{L}(\rho_{AB}) = \text{tr}[\rho_{AB}H] - \mu_1 S(\rho_B) + \mu_2\text{tr}[\rho_{AB}], \qquad (36)$$

where $\mu_i$ are Lagrange multipliers, and we minimise this by setting $\partial_X\mathcal{L}(\rho_{AB}) = 0$, where

$$\partial_X f(\sigma) = \lim_{\varepsilon\to 0}\frac{f(\sigma + \varepsilon X) - f(\sigma)}{\varepsilon}. \qquad (37)$$

To compute the derivative of $S(\rho_B)$, we use

$$\partial_X f(\text{tr}_A[\sigma]) = \partial_{\text{tr}_A[X]}f(\text{tr}_A[\sigma]). \qquad (38)$$

We also use the following formula from ref. [43]

$$\partial_X\log(\sigma) = \int_0^\infty \mathrm{d}u\,\frac{1}{\sigma + u\mathbb{1}}X\frac{1}{\sigma + u\mathbb{1}}, \qquad (39)$$

where $\sigma$ is a density operator. This implies that

$$\text{tr}[\sigma\,\partial_X\log(\sigma)] = \text{tr}[X]. \qquad (40)$$

The result is that we need to find a state $\rho_{AB}$ satisfying

$$\text{tr}\big[X\big(H - \mu_1\mathbb{1}_A\otimes\log(\rho_B) - \mu_1 + \mu_2\big)\big] = 0 \qquad (41)$$

but not for any $X$ because we want to ensure that we only consider pure states. Writing $\rho_{AB} = |\psi\rangle\langle\psi|$, let

$$X = |\phi\rangle\langle\psi| + |\psi\rangle\langle\phi|. \qquad (42)$$

Then, writing $Q = Q^\dagger = H - \mu_1\mathbb{1}_A\otimes\log(\rho_B) - \mu_1 + \mu_2$, we get

$$\langle\psi|Q|\phi\rangle + \langle\phi|Q|\psi\rangle = 0. \qquad (43)$$

But this must be true for any $|\phi\rangle$. Choosing $|\phi\rangle = Q|\psi\rangle$, we get

$$\langle\psi|Q^2|\psi\rangle + \langle\psi|Q^2|\psi\rangle = 0, \qquad (44)$$

which is possible only if $Q|\psi\rangle = 0$. So we have

$$\big[H - \mu_1\mathbb{1}_A\otimes\log(\rho_B) - \mu_1 + \mu_2\big]|\psi\rangle = 0. \qquad (45)$$

Note that $\text{tr}_A[|\psi\rangle\langle\psi|] = \rho_B$, so this is unfortunately not linear.

Equation (45) is difficult to solve in general but may be simplified if we know something about the structure of H. This is the case for the toy model, where H is a sum of commuting terms acting on different pairs of qubits $A_iB_i$. In this case, with the ansatz $|\psi\rangle_{AB} = |\psi_1\rangle_{A_1B_1}\otimes...\otimes|\psi_n\rangle_{A_nB_n}$, we see from Eq. (45) that each $|\psi_i\rangle_{A_iB_i}$ should have the same Schmidt vectors as $|\Phi^+\rangle_{A_iB_i}$. One possible solution is to take all qubit pairs to be in the same state: $|\psi\rangle_{AB} = |\phi\rangle_{A_1B_1}\otimes...\otimes|\phi\rangle_{A_nB_n}$, where $|\phi\rangle = \alpha|00\rangle + \beta|11\rangle$. Then, $S_{\text{initial}} = n$, since the initial state consisted of $n$ EPR pairs, so one need only solve

$$n - m = -n[\alpha^2\log_2(\alpha^2) + \beta^2\log_2(\beta^2)] \qquad (46)$$

for $\alpha$ and $\beta$. And the corresponding energy cost is $\Delta E = n[1-(\alpha + \beta)^2/2]$.

For example, with $m = n/2$, one gets $\Delta E \simeq +0.38\,m$. This is smaller than the optimal energy cost that we saw earlier: $\Delta E = 0.5m$. However, in that case, which was the one-shot protocol, Alice and Bob cannot prepare the state $|\psi\rangle_{AB}$ after extracting $m$ EPR pairs. This is because $|\psi\rangle_{AB}$ has maximal Schmidt rank, so the majorization condition cannot be satisfied, so there is no LOCC process creating $|\psi\rangle_{AB}$.

Interestingly, however, we can create the state $|\psi\rangle_{AB}$ in the asymptotic setting of infinitely many copies of this system, getting a nontrivial upper bound on the optimal energy cost of extracting $m$ EPR pairs per copy of the system. This follows

because, in the asymptotic setting, any bipartite entangled pure state can be transformed into another reversibly using LOCC if their entanglement entropies are the same (see ref. [32] for details of the protocol). So we see that the energy cost of distiling $m$ EPR pairs (per copy of the physical system) in the asymptotic setting will be lower than in the one-shot case. This makes sense: in the asymptotic case, we could just apply any one-shot protocol many times, so the best asymptotic strategy must be at least as good as the best one-shot strategy.

**Method for sampling low energy and low entanglement states**. We now briefly describe a method for sampling states, which, with respect to a given Hamiltonian and bipartition, have low energy and low entanglement. The idea is to start in a random state, and then repeatedly attempt to lower both the energy and entanglement of this state in turn. We represent the state numerically in the form of a Matrix Product State (MPS), and utilise tensor network techniques to lower the energy and entropy.

To lower the energy of the state we perform imaginary time evolution. Specifically we apply an approximation of $e^{-\tau H}$ for some $\tau > 0$, and then renormalise the state, and trim down the bond dimension. We approximate $e^{-\tau H}$ by using a Suzuki–Trotter expansion[44], in a method similar to that used in time-evolving block decimation[45–48].

To lower the entropy we leverage normal forms of MPS. By performing successive singular value decompositions, the Schmidt decomposition of a matrix product state

$$|\psi\rangle = \sum_\alpha \lambda_\alpha|l_\alpha\rangle \otimes |r_\alpha\rangle \qquad (47)$$

can be efficiently calculated[49]. To lower the entropy we want to 'sharpen' the Schmidt spectrum, by raising it to a power

$$|\psi'\rangle \propto \sum_\alpha \lambda_\alpha^{1+\varepsilon}|l_\alpha\rangle \otimes |r_\alpha\rangle, \qquad (48)$$

for some constant $\varepsilon$.

By performing the above procedures successively with random parameters on random states we can sample from states in a manner that is biased towards those with low energy and low entanglement, allowing us to approximate the Pareto front of the two variables as in Fig. 3.

The models we chose to look at are the Heisenberg anti-ferromagnet and the transverse-field Ising model both at criticality:

$$\begin{aligned}H_{\text{HAF}} &= \sum_{n=1}^{N-1}\big(\sigma_n^x\sigma_{n+1}^x + \sigma_n^y\sigma_{n+1}^y + \sigma_n^z\sigma_{n+1}^z\big),\\ H_{\text{TFI}} &= -\sum_{n=1}^{N-1}\sigma_n^z\sigma_{n+1}^z - \sum_{n=1}^{N}\sigma_n^x,\end{aligned} \qquad (49)$$

where $\sigma_n^x$, $\sigma_n^y$ and $\sigma_n^z$ are the Pauli x, y and z matrices.

**Entanglement temperature near ground states of spin chains**. In Fig. 3 we see that $T_{\text{ent}}^A \propto \Delta S$ close to the ground state for the Heisenberg anti-ferromagnet and transverse-field Ising model. This behaviour is in fact generic for quantum spin systems when sufficiently close to the ground state.

Consider starting with the Schmidt decomposition of the ground state $|\Gamma\rangle = \sum_\alpha \lambda_\alpha|l_\alpha\rangle \otimes |r_\alpha\rangle$, and perturbing the highest Schmidt weight

$$|\Gamma'\rangle \propto \sum_\alpha \sqrt{\lambda_\alpha^2 + \varepsilon\delta_{\alpha,0}}|l_\alpha\rangle \otimes |r_\alpha\rangle, \qquad (50)$$

for some $0 \leq \varepsilon \ll \lambda_0$. Taking the Taylor expansions, we find that

$$\begin{aligned}\Delta S &= -\big[S + \log\lambda_0^2\big]\varepsilon + \mathcal{O}(\varepsilon^2),\\ \Delta E &= \Big[\frac{\langle l_0 r_0|H|l_0 r_0\rangle}{4\lambda_0^2}\Big]\varepsilon^2 + \mathcal{O}(\varepsilon^3),\end{aligned} \qquad (51)$$

which indeed implies $T_{\text{ent}}^A \propto \Delta S$ close to the ground state, as observed.

It is worth mentioning that we are primarily interested in the regime of large (but finite) $\Delta S$ extraction, as opposed to $\Delta S \ll 1$ where $T_{\text{ent}}^A \propto \Delta S \simeq 0$. For larger $\Delta S$, we expect that generically $T_{\text{ent}}^A$ is far from zero. In contrast, the small $\Delta S$ regime is analogous to thermodynamics close to absolute zero where the heat capacity vanishes.

**Code availability**. The code used to generate the data is available on request.

## Data availability
The data used to generate the plots in Figs. 3 and 4 are available at ref. [50].

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

## Acknowledgements

We would like to thank David Reeb and Robin Harper for useful discussions. The publication of this article was funded by the Open Access Fund of the Leibniz Association. T.J.O. is supported by the DFG through SFB 1227 (DQ-mat) and the RTG 1991, the ERC grants QFTCMPS and SIQS, and the cluster of excellence EXC201 Quantum Engineering and Space-Time Research. C.T.C. acknowledges support from the ARC via the Centre of Excellence in Engineered Quantum Systems (EQuS), project number CE110001013, and from the AINST Postgraduate Scholarship (John Makepeace Bennett Gift). C.B. was supported by the research fund of Hanyang University (HY-2016-2237).

## Author contributions

All authors were involved in discussions about the work, particularly the toy model and the overall idea. T.F. produced all the QFT results, the bound on the energy cost in terms of the Hamiltonian gap, and wrote most of the manuscript. T.F. and C.T.C. showed that the simple strategy for the toy model is optimal. C.T.C. produced the numerics for spin chains and the argument for linear behaviour of the energy cost at small $\Delta S$. T.J.O. first asked the original question via the toy model and introduced the entanglement temperature. T.J.O., C.B. and T.F. produced the Lagrange multiplier argument.

## Additional information

**Competing interests:** The authors declare no competing interests.

