## [Peer Review File · Nature Communications]

Reviewers' comments:

Reviewer #1 (Remarks to the Author):

The manuscript identifies a very interesting and timely topic, entanglement extraction from the vacuum of complex many-body or field theories, and formalizes its energetic cost as related to the amount of entanglement extracted. Although the setting could seem relatively specific (local non-interacting probes, vacuum state for the system of interest, one-shot protocol, etc), it is of high interest e.g. in the context of quantum field theories and of fundamental value.

They introduce an interesting concept too, the entanglement temperature, which could be potentially related with general-relativistic settings.

The authors introduce an interesting toy model, where some things can be calculated, while at the same time exhibits the difficulty of the problem.

When arguing about extraction cost in general, the authors depict the dependence of entanglement temperature for quantum fields in different dimensions. In this section, however, it is not very clear to me to what extent the results are new or the authors simply restate known results from the literature. In any case it would very much help if they were to specify which quantum fields they are talking about and the range of validity of their claims (free fields?, massless?, etc).

Next, they introduce three methods to tackle this very hard problem:

__method 1__: this method uses Lagrange multipliers to state the final AB state that can minimize energy cost for a given entanglement extraction (m EPR pairs). They show that for the toy model the energy can be minimized below the simple protocol given one page before: $\Delta E = 0.5m$. From here on, e.g. the sentence "However, in the one-shot setting, Alice and Bob cannot prepare the state $|\psi\rangle_{AB}$ after extracting m EPR pairs (because $|\psi\rangle_{AB}$ has maximal Schmidt rank)", the exposition should be improved: what protocol gets you to the state $|\psi\rangle_{AB}$, etc.

__method 2__: this method uses the distance (overlap) between the final entanglement-extracted state and the vacuum state as the minimization target. It is an interesting idea, but one wonders whether the authors could say something more about which kind of models would be best suited to this method (models where energetic distance is related with overlap distance from the ground state?).

__method 3__: the last method consists in sampling the Hilbert space representing states as matrix products, with entropy- and energy-lowering techniques. From this the asymptotic temperature is obtained. For the chosen models, at criticality, the temperature seems to diverge. The authors should comment a bit more on this point, and its relation to criticality.

I think the paper should be accepted, but with some better exposition of the why this is important, its relation with possible general-relativistic vs q. info. topics, etc. Also, even if this manuscript is intended for quantum information researchers, it would do no harm to clarify some mathematical steps for the more general reader.

Some further possible improvements:

- as said above, what is the interest of this protocol? is it entanglement harvesting from QFT with the less energetic disturbance? Do we really learn something about a QFT or many-body system with these proposed quantities?
- the authors should not avoid stressing more "extraction FROM THE VACUUM". It is of sufficient interest and generality. And this leads to the next point
- Is it interesting to use these quantities to look at excited states of QFT or many-body theories? What would be the interest/difficulties?

In summary, I would say that the paper, which is very good in its proposed ideas and results, would benefit mostly from a better outlook of the relation of these concepts to modern problems in related fields.

This work could influence many different communities, from pure quantum information, to many-body states representation, to energy harvesting in flat-space QFT, to problems in QFT in curved space-time, etc.

It is also technically sound and its results can be tracked correctly from the article exposition.

Reviewer #2 (Remarks to the Author):

In this paper the authors discuss the issue of the energy cost for extracting quantum entanglement from a complex quantum system, subject to an energy constraint. The main area of application is quantum field theory and condensed matter, in particular in the one shot regime, namely when only one copy of the state of the system is available.

A protocol for the entanglement extraction is developed and applied to a toy model, for which an exact solution can be achieved.

Having defined a procedure for quantifying the energy amount necessary for extracting quantum entanglement (which is characterized by an entropy), makes possible the introduction of an entanglement temperature in a very straightforward and natural way, in analogy with classical thermodynamics.

They then present some bounds and numerical techniques that allow to estimate it for the case of interest presented in the text.

The scaling relations between energy cost and entanglement entropy are studied for systems in different dimensions, exhibiting a strong dependence on the dimensionality.

The paper is very well written and represents an important contribution to the field, with possible applications and further conceptual developments. I recommend it for publication in Nature Communications.

PS

There is a typing error in the formula (22): instead of $\langle \psi | \psi \rangle$, it should be $\langle \psi | \psi \rangle$ for the two last terms.

Reviewer #3 (Remarks to the Author):

This paper examines the energy cost of entanglement extraction via LOCC and how the optimal energy cost scales with the number of extracted EPR pairs. The entanglement temperature is defined to relate the energy cost to the amount of extracted entanglement. It is shown that in one dimension the energy cost grows exponentially with the number of extracted EPR pairs. The antiferromagnetic Heisenberg model and the transverse Ising model are numerically investigated to calculate the entanglement temperature and bound the energy cost of entanglement extraction using matrix product states.

This is an interesting problem which is relevant to quantum control. However, in the manuscript the energy cost of a quantum operation is defined rather than determined from physics principles. Thus I cannot judge whether or not the energy cost of extracted EPR pair which is discussed here

is of fundamental importance. In more detail, the energy cost is defined here as the difference between the initial and final states of the system from which EPR pairs are extracted. However, no energy cost needed to perform operations for this task is considered. The authors simply assume that "Alice and Bob can carry out any local operation they like on the ancillary degrees of freedom with no energy cost." Physically and also practically, the most challenging task always lies in these operations and without serious investigation into this problem one can tell only the part of the energy cost of our concern. Thus the energy cost discussed here only gives a rough lower bound for any real or thought experiment.

Another major claim in the manuscript is the cooling of the system by entanglement extraction. However, what is actually shown is that the energy of the system may go down, whereas no discussions on thermodynamic temperature are made. The entanglement temperature defined in (13) is also confusing because no thermodynamic argument is provided here.

On technical sides, the authors argue below (11) that the simple protocol given there is optimal. However, I see no rigorous (analytic) argument in support of this. Only a numerical piece of evidence is provided. One might also easily guess from (14) the relationship between the entanglement entropy and the energy scale $1/a$ on the basis of conformal field theory. So what is new about this?

Some minor points. Ent in (5) is not defined. It is unclear whether the definition of the energy cost of entanglement extraction defined below (5) is the same as that given in the figure caption of Fig. 1.

All in all, I feel that this manuscript is constituted from a collection of small ideas and somewhat sketchy arguments but no serious proof or a substantiated ground-breaking idea is given. I therefore cannot recommend the present manuscript for publication in Nature Communications.

REPLY TO REVIEWERS

Authors: We thank each of the reviewers for taking the time to review this work and for providing insightful comments. We address each of the reviewers' points below and describe the changes made to the manuscript based on their recommendations. Additionally, we have corrected one or two typos and did some minor editing. The largest changes appear in (i) the section “Extracting entanglement subject to an energy constraint”, where we have added a more comprehensive account of the energy cost arising during an entanglement extraction protocol and (ii) the section “The energy cost in general”, where we have improved the derivation of the results, and emphasized what is new and what is from the literature. We discuss these changes in more detail and address the other points raised by the reviewers below.

(The quoted text below from the reviewers is identical to their reports, except that mathematics has put into latex form. Changes to the draft based off of comments from reviewers 1, 2 and 3 are coloured in red, green and blue respectively, while changes resulting from comments by more than one reviewer are in orange. Additionally, we fixed one or two typos, or made small additional changes, which are coloured in magenta.)

Reviewer 1

The manuscript identifies a very interesting and timely topic, entanglement extraction from the vacuum of complex many-body or field theories, and formalizes its energetic cost as related to the amount of entanglement extracted. Although the setting could seem relatively specific (local non-interacting probes, vacuum state for the system of interest, one-shot protocol, etc), it is of high interest e.g. in the context of quantum field theories and of fundamental value. They introduce an interesting concept too, the entanglement temperature, which could be potentially related with general-relativistic settings. The authors introduce an interesting toy model, where some things can be calculated, while at the same time exhibits the difficulty of the problem. When arguing about extraction cost in general, the authors depict the dependence of entanglement temperature for quantum fields in different dimensions. In this section, however, it is not very clear to me to what extent the results are new or the authors simply restate known results from the literature. In any case it would very much help if they were to specify which quantum fields they are talking about and the range of validity of their claims (free fields?, massless?, etc).

Authors: We have rewritten much of this section (“The energy cost in general”) to make the derivation, as well as what is new, clearer. The entanglement entropy formulas for ground states of lattice systems are taken from the literature. By estimating the energy cost of extracting all the entanglement from a lattice quantum field theory vacuum with lattice spacing a , we get the energy cost, which is a function of a . Then we combine this with the entropy formulas to get a lower bound on how much energy one needs to extract m EPR pairs from the vacuum.

We have also written more about the range of applicability of the results in this section. Basically, we can get a physical estimate on the energy cost of extracting entanglement for any QFT where the entanglement entropy can be calculated (for a lattice regularization). This includes free massive bosonic QFTs, but it is also believed that more general models should obey an area law for the entanglement entropy, which would allow us to generalize the results. (Of course, we just give a rough physical argument. In principle, it may be possible to obtain precise formulas for specific QFTs.)

Next, they introduce three methods to tackle this very hard problem:

__method 1__.: this method uses Lagrange multipliers to state the final AB state that can minimize energy cost for a given entanglement extraction (m EPR pairs). They show that for the toy model the energy can be minimized below the simple protocol given one page before: $\Delta E = 0.5m$. From here on, e.g. the sentence “However, in the one-shot setting, Alice and Bob cannot prepare the state $|\psi\rangle_{AB}$ after extracting m EPR pairs (because $|\psi\rangle_{AB}$ has maximal Schmidt rank)”, the exposition should be improved: what protocol gets you to the state $|\psi\rangle_{AB}$, etc.

Authors: We have expanded the explanation in the section “Method I” to make things clearer. In both the asymptotic and one-shot case, the protocols converting between pure states are unfortunately a little complicated, so we refer to a standard reference.

__method 2__.: this method uses the distance (overlap) between the final entanglement-extracted state and the vacuum state as the minimization target. It is an interesting idea, but one wonders whether the authors could say something more about which kind of models would be best suited to this method (models where energetic distance is related with overlap distance from the ground state?).

Authors: We have added a more concrete physical application of this method to the section “Method II”. This allows one to lower bound the energy cost in terms of the Hamiltonian’s gap (if it has one) and the overlap of the final state with the ground state. (We feel that this method has become far more interesting after following this suggestion.)

__method 3__.: the last method consists in sampling the Hilbert space representing states as matrix products, with entropy- and energy-lowering techniques. From this the asymptotic temperature is obtained. For the chosen models, at criticality, the temperature seems to diverge. The authors should comment a bit more on this point, and its relation to criticality.

Authors: In this section “Method III”, we have added a discussion of the behaviour of these plots and how they relate to the fact that the models studied are critical. Actually, the behaviour that was most easy to address was why the entanglement temperature

decreases with the system size (for a fixed amount of extracted entanglement).

Interestingly, the fact that the entanglement temperature appears to diverge seems to be a finite-size effect. The total amount of extractable entanglement grows with the system size, and the energy cost of extracting all the entanglement is bounded above (by the coupling between Alice and Bob's systems, which is just the coupling between two halves of the spin chain). Therefore, the maximum entanglement temperature will be bounded (since $T_{\text{ent}} = \Delta E / \Delta S$). We have also pointed this out in the same section.

I think the paper should be accepted, but with some better exposition of the why this is important, its relation with possible general-relativistic vs q. info. topics, etc. Also, even if this manuscript is intended for quantum information researchers, it would do no harm to clarify some mathematical steps for the more general reader.

Authors: We have added more detail about importance in both the introduction and the abstract, and we have highlighted the relation with gravity in the introduction too.

We have also made efforts to elucidate many of the steps where we used quantum information techniques that may not be familiar to physicists from other fields. Locations where this is done are (i) in "Preliminaries", lines 124 – 127, (ii) same section, lines 137 – 142, (iii), in "Extracting entanglement subject to an energy constraint" lines 163 – 167, and (iv) in "A toy model", lines 317 – 325.

Some further possible improvements:

- as said above, what is the interest of this protocol? is it entanglement harvesting from QFT with the less energetic disturbance? Do we really learn something about a QFT or many-body system with these proposed quantities?

Authors: Entanglement harvesting is certainly one setting where these results are useful, as the energy cost of entanglement harvesting is lower bounded. In principle, this should be useful for inspiring energy-efficient protocols for extracting entanglement.

We also learn something about QFTs or many-body systems. In fact, the motivation of this work was exactly this question: how is entanglement organized in highly entangled states of complex many-body systems? One way to attack this is to ask the question from a practical point of view. If we have access to some amount of energy (one resource) how much entanglement can we trade it for (another resource). We have added some text elaborating on the answers to these questions in the abstract and introduction. (This overlaps somewhat with the text added in response to the previous point in the abstract and the introduction.)

- the authors should not avoid stressing more "extraction FROM THE VACUUM". It is of sufficient interest and generality. And this leads to the next point

Authors: We have pointed this out a few more times (e.g., in the abstract and the introduction).

- Is it interesting to use these quantities to look at excited states of QFT or many-body theories? What would be the interest/difficulties?

Authors: This is an interesting question because excited states have a high amount of entanglement (often obeying a volume law). One might also consider mixed entangled states, particularly thermal states at sufficiently low temperatures. We have mentioned these questions in the outlook, as they would form the basis for some interesting future work. Since thermal states are mixed at non zero temperature, quantifying entanglement becomes much harder. On the other hand, excited states would be pure, so quantifying entanglement would be easier. Actually, in that case it is conceivable that Alice and Bob could extract entanglement while also extracting energy from the system, though this may not be possible with LOCC.

In summary, I would say that the paper, which is very good in its proposed ideas and results, would benefit mostly from a better outlook of the relation of these concepts to modern problems in related fields. This work could influence many different communities, from pure quantum information, to many-body states representation, to energy harvesting in flat-space QFT, to problems in QFT in curved space-time, etc.

It is also technically sound and its results can be tracked correctly from the article exposition.

Authors: We thank the reviewer again for the positive feedback and for many detailed suggestions.

Reviewer 2

In this paper the authors discuss the issue of the energy cost for extracting quantum entanglement from a complex quantum system, subject to an energy constraint. The main area of application is quantum field theory and condensed matter, in particular in the one shot regime, namely when only one copy of the state of the system is available.

A protocol for the entanglement extraction is developed and applied to a toy model, for which an exact solution can be achieved.

Having defined a procedure for quantifying the energy amount necessary for extracting quantum entanglement (which is characterized by an entropy), makes possible the introduction of an entanglement temperature in a very straightforward

ward and natural way, in analogy with classical thermodynamics.

They then present some bounds and numerical techniques that allow to estimate it for the case of interest presented in the text.

The scaling relations between energy cost and entanglement entropy are studied for systems in different dimensions, exhibiting a strong dependence on the dimensionality.

The paper is very well written and represents an important contribution to the field, with possible applications and further conceptual developments. I recommend it for publication in Nature Communications.

Authors: We thank the reviewer for taking the time to review the paper and for the positive feedback.

PS There is a typing error in the formula (22): instead of $\psi^\dagger\psi$, it should be $\psi\psi^\dagger$ for the two last terms.

Authors: Good catch! We have fixed this typo.

Reviewer 3

This paper examines the energy cost of entanglement extraction via LOCC and how the optimal energy cost scales with the number of extracted EPR pairs. The entanglement temperature is defined to relate the energy cost to the amount of extracted entanglement. It is shown that in one dimension the energy cost grows exponentially with the number of extracted EPR pairs. The antiferromagnetic Heisenberg model and the transverse Ising model are numerically investigated to calculate the entanglement temperature and bound the energy cost of entanglement extraction using matrix product states.

This is an interesting problem which is relevant to quantum control. However, in the manuscript the energy cost of a quantum operation is defined rather than determined from physics principles. Thus I cannot judge whether or not the energy cost of extracted EPR pair which is discussed here is of fundamental importance. In more detail, the energy cost is defined here as the difference between the initial and final states of the system from which EPR pairs are extracted. However, no energy cost needed to perform operations for this task is considered. The authors simply assume that “Alice and Bob can carry out any local operation they like on the ancillary degrees of freedom with no energy cost.” Physically and also practically, the most challenging task always lies in these operations and without serious investigation into this problem one can tell only the part of the energy cost of our concern. Thus the energy cost discussed here only gives a rough lower bound for any real or thought experiment.

Authors: The reviewer raises a very good point, and we have added a discussion of this in the section “Extracting entanglement subject to an energy constraint”. (We feel that the exposition has benefited greatly from this addition.) The new section shows that the energy

costs we discuss are of fundamental importance, as we get a lower bound on the energy Alice and Bob need to extract the entanglement (if their ancillas have some nontrivial Hamiltonians, this can easily be taken into account).

It is true that we give a lower bound, but any additional energy costs would depend strongly on the setup, whereas our lower bound is independent of any specific protocol. Also, it is worth mentioning that even such theoretical lower bounds can be extremely useful. For example, Landauer’s erasure principle gives a theoretical lower bound on the amount of energy needed to erase information (usually in the context of computation), and it has been of fundamental importance in research despite the fact that current technology is nowhere near saturating the Landauer bound (in other words, erasing information is much less efficient in practice). The reviewer is, of course, correct that precise calculations for energy costs are highly useful, but we believe that theoretical lower bounds are quite important too.

There is also a second point, which was one of the motivations for this work. Quantifying the minimum energy needed to extract different amounts of entanglement gives us some insight into the structure of entanglement in complex systems. In this context, it makes sense to define the energy cost independently of the details of any apparatus used by Alice and Bob.

Another major claim in the manuscript is the cooling of the system by entanglement extraction. However, what is actually shown is that the energy of the system may go down, whereas no discussions on thermodynamic temperature are made. The entanglement temperature defined in (13) is also confusing because no thermodynamic argument is provided here.

Authors: We have rephrased that statement in the section “Extracting entanglement subject to an energy constraint” to avoid any confusion (lines 185 – 188). The reviewer is indeed correct that in that context we are not dealing with thermal states (except the vacuum), and so it does not make sense to mention “cooling”. Whether there are cases where Alice and Bob can lower the energy of the physical system (possibly extracting and later using this energy as a resource) while extracting entanglement was not meant to be one of our main claims. We were just pointing out that it may be possible in some situations.

We have written more in the section “The entanglement temperature” to highlight that it is not related to thermodynamic temperature, only defined by analogy.

On technical sides, the authors argue below (11) that the simple protocol given there is optimal. However, I see no rigorous (analytic) argument in support of this. Only a numerical piece of evidence is provided. One might also easily

guess from (14) the relationship between the entanglement entropy and the energy scale $1/a$ on the basis of conformal field theory. So what is new about this?

Authors: Regarding the first point, the reviewer is correct that the argument is not rigorous. Instead, it is a combination of numerics and necessary criteria for pure state transformations using LOCC. We have made this more clear in section “A toy model” (lines 312 – 314).

Regarding the second point, we agree with the reviewer that the derivation here (which uses known results on the entanglement entropy for lattice models) is not particularly complex, but we believe it is of physical interest. Furthermore, it is a new result that the energy cost of extracting m EPR pairs grows *at least* exponentially in m for massive free quantum fields in $(1+1)$ dimensions (we bound the energy cost for free models in higher dimensions too). We believe this is quite interesting, and it opens the door to further refinements, which would have applications to, e.g., more specialized entanglement harvesting protocols. We have rewritten parts of this section “The energy cost in general” to highlight what is new.

Some minor points. Ent in (5) is not defined. It is unclear whether the definition of the energy cost of entanglement extraction defined below (5) is the same as that given in the figure caption of Fig. 1.

Authors: We wanted to leave the definition of Ent flexible because there are many possible entanglement measures with different applications. But this was indeed vague, so we have added a few sentences in “Extracting entanglement subject to an energy constraint” (lines 173–184) to discuss this and point out that we use the entanglement entropy in the rest of the paper.

The reviewer is correct that these definitions of energy cost do not agree, and we have amended the caption of Figure 1 to rectify this.

All in all, I feel that this manuscript is constituted from a collection of small ideas and somewhat sketchy arguments but no serious proof or a substantiated ground-breaking idea is given. I therefore cannot recommend the present manuscript for publication in Nature Communications.

Authors: We thank the reviewer for taking the time to review this paper and for their detailed comments and suggestions.

We should add, however, that there is indeed an overarching substantial idea, which is new and physically motivated: the connection of two resources, entanglement and the energy needed to access it in a useful way. (Frankly, it came as a surprise to us that there previously existed no general theory doing this.)

On the one hand, this has immediate applications to, e.g., bounding the efficiency of entanglement harvesting. On the other hand, we believe one can learn about the entanglement structure of complicated many-body states with this framework, as this gives a new perspective: how much useful entanglement does a state contain and how much energy does it take to access it. This may become even more interesting for thermal states where (because one is dealing with mixed entangled states) it is hard to separate out how much entanglement is present and useful (i.e., extractable). Our framework is operationally motivated: calculating, e.g., Renyi entanglement entropies for ground states is interesting in its own right, but connecting the amount of entanglement present to something physically motivated (the energy needed to extract it) gives one a different point of view.

REVIEWERS' COMMENTS:

Reviewer #1 (Remarks to the Author):

The points raised have been satisfactorily addressed by the authors.

Reviewer #3 (Remarks to the Author):

I have read the revised manuscript and the authors' reply to the first round of referee reports and found that the authors have responded to my criticisms reasonably well by making it clear what they meant in their original claims such as the entanglement temperature. It now becomes clear what it means by extracting EPR pairs from quantum field vacua. While it seems unclear how to experimentally test the proposed idea, the results of toy model calculations such as an exponential increase of energy cost of entanglement extraction are very interesting. I recommend the publication of the manuscript in Nature Communications.